# An Overview of Recent Progress in Engineering Three-Dimensional Scaffolds for Cultured Meat Production

**DOI:** 10.3390/foods12132614

**Published:** 2023-07-06

**Authors:** Yuan Wang, Liqiang Zou, Wei Liu, Xing Chen

**Affiliations:** 1State Key Laboratory of Food Science and Resources, College of Food Science & Technology, Nanchang University, Nanchang 330047, China; 2National R&D Center for Freshwater Fish Processing, Jiangxi Normal University, Nanchang 330022, China; 3School of Life Sciences, Nanchang University, Nanchang 330031, China

**Keywords:** cultured meat, biomaterials, three-dimensional cultivation technology, scaffolding construction

## Abstract

Cultured meat is a new type of green, safe, healthy, and sustainable alternative to traditional meat that will potentially alleviate the environmental impact of animal farming and reduce the requirement for animal slaughter. However, the cultured meat structures that have been prepared lack sufficient tissue alignment. To create a product that is similar in texture and taste to traditional animal meat, muscle stem cells must be organized in a way that imitates the natural structure of animal tissue. Recently, various scaffold technologies and biomaterials have been developed to support the three-dimensional (3D) cultivation and organization of muscle stem cells. Hence, we propose an overview of the latest advancements and challenges in creating three-dimensional scaffolds for the biomanufacturing of cultured meat.

## 1. Introduction

The sustained surge in global production and consumption demand of meat is attributable to the improvement in people’s living standards and the growing population of the world [1]. The Food and Agriculture Organization of the United Nations (FAO) has forecasted that the global population may reach 9.7 billion by 2050 and the global demand for meat is estimated to reach 455 million tons [2,3]. However, traditional animal husbandry to meet large-scale demands pose issues related to environmental resources, public health, and animal welfare [4]. Livestock production involves 30% of the world’s land resources, consumes more than 8% of human water consumption, and contributes up to 18% of greenhouse gas emissions in CO_2_ equivalent [5]. Just producing 1 kg of beef requires 40 m^2^ of land, 15 m^3^ of water, and 300 kg of CO_2_ equivalent [6]. Additionally, over 75% of infectious diseases in humans come from animals (zoonotic), and the growing trend in vegetarians and animal rights advocates tends to reduce consumption of livestock products [7,8]. To fulfil the increased food demands of the growing population of the world using limited resources in our planet, humanity is searching for safer and more sustainable strategies [3]. Therefore, emerging are green, healthy, safe, and sustainable meat substitutes, such as plant-based meat and cultured meat, to alleviate the various problematic factors associated with traditional meat production and explore a new pathway to a sustainable ecosystem.

The term “cultured meat”, also known as lab-grown meat or in vitro meat, refers to edible artificial meat tissue made by saving stem cells from different animals (pork, chicken, and beef) without the livestock farming treatment [4]. The four major components of bioartificial meat include isolating cell tissues, expanding cells in bioreactors, culturing cells on specific scaffolds, and further processing to create the final product [9]. Two prominent results for this final product are to accurately mimic sensations of real meat and ensure positive consumer acceptance [10,11]. Besides these strategies, Tuomisto found that cultured meat can effectively protect the environment by reducing energy use (7–45%), greenhouse gas emissions (78–96%), land use (99%), and water resource utilization (82–96%), and still avoid various concerns caused by traditional husbandry such as neglecting animal welfare and abusing antibiotics [12]. Accordingly, cultured meat may be advertised as an environmentally friendly meat alternative that closely resembles real meat.

Mimicking the sensations of livestock meat can be categorized in appearance, texture, taste, and nutritional value. Additionally, cost and mass must be taken into consideration [13]. One key aspect is texture, which should possess a dense, fibrous structure and good elasticity [14]. The reason is that meat is mainly composed of skeletal muscle tissue; myoblasts transform into mature muscle fiber bundles and tissues with specific length and thickness after being incubated [10]. Popular properties such as elasticity are beneficial for consumers and provide a more authentic meat-eating experience [14]. However, meat analogues currently described using extrusion, 3D printing, and microcarriers through cultivating stem cells present challenges due to their loose structure and lack of chewiness. Hence, to fabricate anisotropic meat products with fibrous morphology, cells need to be aligned and grown in a single direction using scaffolds that enhance the sensory and quality experience for consumers. Likewise, culture scaffolds should replicate the natural extracellular matrix (ECM) of cells while also ensuring ample space for air and nutrient exchange and waste removal, as well as utilizing edible and cost-effective biomaterials [15]. This is where 3D scaffolds technologies come in, providing a framework for the cell growth and high-grade cultured meat production.

Furthermore, the level of consumer acceptance of cultured meat is a crucial factor that influences the commercial viability of the market [16]. According to academic ethics communities, cultured meat is currently considered morally permissible [17]. Surveys conducted in the United States and Italy showed that 2/3rds and 54% of respondents, respectively, were willing to try it [18,19]. Additionally, vegetarians believe that cultured meat is morally justifiable and could serve as a healthy substitute for meat [8]. Therefore, cultured meat will be widely accepted by consumers and possess enormous potential for development and market prospects.

This article overviews the development history of cultured meat and outlines the different scaffold technologies and biomaterials while expounding the advantages and challenges in the food field. Finally, the prospects of this article are analyzed comprehensively and proposed in terms of cultivating meat.

## 2. Historical Perspective of Cultured Meat

Cell culture is the course of dispersing a piece of tissue into single cells originating from a bio-organ or directly extracting single cells from the organism, and also involves scattering in vitro cells under extracellular conditions and allowing them to survive and proliferate. The earliest cell culture was conducted in 1885 by German scientist Roux, who used mild physiological saline to cultivate chicken embryo tissue that could live for several months [20]. In 1907, Harrison, an American embryologist, established an in vitro cultivation method which successfully cultured frog embryo neurons in a sterile environment [21]. Additionally, this experiment has been recognized as the true inception of cell tissue culture, offers a reproducible technique, and showcases the fact that cells are capable of preserving their biological functions in vitro [20]. Subsequently, an increasing number of scientists have turned their attention to in vitro cell growth research, which provides a strong groundwork for animal tissue engineering. Nowadays, animal tissue engineering based on cell culture is widely developed for use in biomedical fields [22]. This has resulted in the accumulation of valuable experience for the manufacture of lab-grown meat, further propelling its development.

In practice, cultured meat, as a type of animal food, is a novel technological innovation that merges animal tissue engineering with edible biomaterials. The first vision of cultured meat was reported in 1931 by Churchill: “We shall escape the absurdity of growing a whole chicken in order to eat the breast or wing, by growing these parts separately under a suitable medium. Synthetic food will, of course, also be used in the future” [23]. In 2002, Morris Benjaminson successfully conducted early cultured meat experiments. He incubated edible muscle protein from goldfish cells, and the project was sponsored by the National Aeronautics and Space Administration (NASA) [24]. After over 80 years of work, Churchill’s vision has been achieved. The world’s first cell-cultured meat was unveiled in 2013 by Mark Post, a member of a laboratory at Maastricht University in the Netherlands. Post spent two years and USD 280,000 to complete the creation of muscle strip beef hamburgers, which was an edible product [25]. Selling cultured chicken was developed in America by Upside Foods Company in 2015. Equally, China’s first product occurred on 18 November 2019. Zhou Guanghong, a professor at Nanjing Agricultural University, successfully cultivated the sixth generation of pig muscle stem cells in a nutrient solution over 20 days, resulting in a 5 g meat product [26]. Based on this study, Nanjing Zhouzi Future Food Technology Co., Ltd., (Nanjing, China) established the first domestic production platform for “cultured meat” in the same year. The first commercialization of cultured meat was established by the enterprise Future Meat in Israel, with a daily production capacity of 500 kg of products, introduced after 2020. Nowadays, cultured meat engineering is hopeful for commercialization in the future. Figure 1 summarizes the development flow chart of cultured meat.

## 3. Scaffolding Biomaterials for Cultured Meat

Scaffolding biomaterials have impacts on the structure and properties of scaffolds. The features of scaffolding biomaterials typically possess excellent biocompatibility, high porosity, and the ability to restore the ECM, while ensuring sufficient mechanical strength to guide cell adhesion, proliferation, and morphological changes [27]. According to policy made by the Food Safety Supervision Bureau, the most critical attributes for biomaterials used in cultured meat construction include edibility, non-animal origin, sustainability, and commercial viability. Other essential factors to consider involve biodegradability and restrictions on the presence of non-edible and/or toxic compounds, such as solvents and crosslinking agents [9].

Proteoglycans, collagen, and glycoproteins make up the natural ECM; therefore, proteins and polysaccharides are supposed to be the main elements of scaffold biomaterials [15,28]. Proteins can be divided into animal protein, plant protein, and fungal protein, and polysaccharides consist of plant and animal polysaccharides. Some plant polysaccharides, such as alginate, pectin, konjac gum, and cellulose, have potential as useful biomaterials because of their physiological functions and excellent cellular adhesion [9]. However, due to sensitivity towards animal welfare and sustainability issues, animal polysaccharides are not recommended [9]. Briefly, animal-derived, plant-derived, and synthetic polymer biomaterials are the main groups of scaffold biomaterials.

### 3.1. Animal-Derived Biomaterials

Biomaterials derived from animal are rich in ECM and promote better cellular growth, with the added benefit of being fully absorbed by the human body; for instance, elastin, gelatin, collagen, and fibronectin [29]. It is expected that these materials combined are very similar to conventional meat in structure. Enrione J et al. prepared an edible porous scaffold through freeze-drying technology which incorporated salmon gelatin, agar, and sodium alginate. This scaffold allowed muscle stem cells to adhere and grow, resulting in suitable myogenic responses [30]. Another example involved applying electrospinning techniques in combination with porcine gelatin and TG enzyme, alongside chemical crosslinking agents. The micro gelatin fibers obtained were conducive to the growth of muscle cells and facilitated their unidirectional alignment [31]. However, scaffolds of a single material have inferior mechanical properties, and some collagen (from fish skin) and gelatin (from pigs and cowhide) materials are expensive and susceptible to environmental and ethical issues, making them difficult to be accepted sustainability-wise [32,33].

To fully explore natural protein resources, non-traditional animal proteins have attracted interest in the food industry. Edible insects, in particular, have been garnering attention due to their rich protein and fat content, as well as their more nutritious and sustainable value [34]. Furthermore, depending on the hotpot of non-animal protein substances, collagen and gelatin extracted from plants, yeast, and bacteria can support cell adhesion and arranged growth [32]. A recent study found that some edible fungi, such as enoki mushroom polysaccharides in the shape of natural fiber, had multiple biological activities and promising biomaterials to fabricate scaffolds, but there are currently no relevant reports [35]. Consequently, preparing scaffolds using natural protein materials is an area that requires further exploration in lab-grown meat.

### 3.2. Plant-Derived Biomaterials

Plant proteins and natural plant tissues may hold promise to become plant-based biomaterials. Plant-based proteins are the prime option for growing meat biomaterials owing to their nutritional value, low cost, excellent cellular compatibility, and perfect consumer acceptance [36]. For example, soy protein isolate (SPI), a type of high-quality plant protein, is abundant in essential amino acids and various vitamins. It is also very suitable for cell attachment, and includes some cereal proteins [37,38]. Textured soy protein scaffolds [13] and 3D fiber-alignment scaffolds from wheat gluten [39] have been investigated for their potential in developing cultured meat. The adhesion and proliferation of pig smooth muscle cells are perfectly reflected on the peanut drawn protein scaffold, and the cell survival rate is notably high. The produced ECM protein and muscle protein endow the final product with excellent quality [40]. Thus, it is a promising approach that replacing animal protein with plant protein settles the nutritional and health problems of traditional animal protein.

Natural plant tissues, like green and edible spinach leaves, celery, and apple tissue [41], show obvious vascularity and porous morphology (anisotropic structure) to facilitate oxygen and nutrient transport, serving a favorable cell environmental platform [15]. Allan et al. demonstrated that a decellularized grass leaf scaffold with natural morphology supported the attachment and proliferation of mouse C2C12 cells and induced the cells’ alignment [42]. These results indicate that using plant sources as scaffold materials is feasible, to some extent, and provides a theoretical basis. However, it is possible to note the prospective issues with taste, scalability, and tissue applicability in the future.

### 3.3. Synthetic Polymer Biomaterials

Currently, synthetic polymer scaffolding materials for tissue or pharmaceutical engineering are typically designed as microcarriers or other forms to make up porous, fibrous, and anisotropic structures [29]. They have distinct components that provide adjustability, mechanical behavior, and biocompatibility, such as polyvinyl alcohol, polypropylene, polycaprolactone, and polylactic acid [15]. Kankala et al. used a porous microcarrier composed of a poly(lactic-co-glycolic acid) and gelatin to support the high adhesion rate of C2C12 myoblast cells and their tight fiber-like alignment, offering a fresh method of muscle regeneration [43]. A double-layer scaffold based on a mixture of uniaxially aligned PCL fibers and anisotropic methacrylic acid alginate was shown to gain muscular tubes. These tubes contracted upon electrical stimulation, expanding the application of vascularized tissues [44].

Many biomaterials in the food industry are limited because of they are non-edible and susceptible to degradation, which could cause damage to tissues [15]. Fortunately, according to the General Standard for Food Additives, polyethylene glycol has been approved for addition with a maximum allowable amount of 1–70 g/kg [45]. Hence, to determine its suitability as a scaffold material, concentration levels may play a key role; more importantly, we must focus on expediting the degradation rate and addressing food safety concerns. In short, ensuring the edibility of synthetic polymer materials is a huge challenge. Regarding the various biomaterials mentioned above, we summarize their advantages and disadvantages in Table 1.

## 4. Three-Dimensional Scaffold Technologies

During the past few years, numerous new technologies have emerged and been utilized in the field of engineering three-dimensional scaffolds, notably, technologies such as 3D printing, electrospinning, extrusion, directional freezing, electric field, cell microcarriers, plant tissue decellularization, and cell sheet (Table 2). These technologies have garnered significant attention as they hold promising potential for the efficient production of cultured meat.

### 4.1. Three-Dimensional Printing

Three-dimensional printing, also known as additive manufacturing, is generally defined as the process of printing functional materials using 3D model computer data via layer-by-layer self-assembly technology. This process can generate biologically similar structures to natural materials [57]. Since its inception, continuous progression has made this technology flourish in tissue engineering, manufacturing engineering, and biomedicine (disease modeling, prosthetics, and cell culture) [58,59]. Current bioprinting techniques include inkjet-based bioprinting, laser-assisted bioprinting, stereolithography-based bioprinting, and extrusion-based bioprinting [60]. Only extrusion-based bioprinting has high correlation to constructing cultured meat scaffolds, with the ability to simulate natural tissues, and the progress is simple and convenient [60]. Other methods involving biomaterials have drawbacks such as food incompatibility, metal residue, toxicity and carcinogenicity, high cost, and strong dependence on devices [9].

A common printing method in cell engineering is bio-ink that consists of cells, basic materials, and other necessary components [61], and it must be edible or completely degradative in cultured meat manufacturing [15]. Shulamit’s group concluded that bovine satellite cells (BSCs) successfully simulated muscle growth, resulting in the achievement of natural meat thickness on a multi-layered fiber network scaffold [46]. The scaffold was printed using extrusion-based 3D printing technology in a gelatin support bath, and employing a combination of pea protein isolate, soy protein isolate, and seaweed salts as raw materials (Figure 2a). Kang et al. used tendon-gel integrated bioprinting (TIP) to create gelatin fibers and beef steak tissue, which cultured both BSCs and adipose-derived stem cells to form muscle, fat, and blood vessel fibers, resulting in the possibility of constructing meat with a certain fiber content [47].

### 4.2. Electrospinning

Electrospinning is a simple, economical, and adjustable technology for manufacturing fibers [31]. Generally, a solution containing biomaterials is injected through an injector needle under a strong electric field, and the droplets are transformed from spherical to conical to form a Taylor cone. As the solvent evaporates and becomes thinner, the conical droplets solidify and become fiber filaments, which continuously deposit on a grounded collector. The obtained fiber filaments are solid micro- or nanometer sizes [62]. This technology has been well demonstrated in the textile industry, nano-equipment, and organizational engineering, as well as industrial production [31,63].

Regarding cell cultivation, there are two vital parts to consider. Firstly, cells can attach to the fibrils of different biomaterials, and myogenesis occurs via constructing scaffolds to induce cell alignment and promote muscle cell growth [64,65]. Secondly, the high specific surface area of the nanofiber scaffold can provide additional attachment points and precise structural remodeling for cell adhesion and proliferation [66]. Inspired by natural sponges, an electrospun short, fibrous sponge with 3D morphology and biomimetic characteristics similar to ECM, constructed from gelatin and polylactic acid as raw materials as biomaterials, can provide a favorable living environment for HUVECs and facilitate the 3D regeneration of cells and blood vessels, making it a promising candidate for skin tissue repair and other medical applications [67]. In the food field, collagen, gelatin, whey protein, chitosan, cellulose, and starch are edible materials that can be considered [9]. Luke A. MacQueen et al. obtained meat-like products by culturing rabbit skeletal muscle cells and bovine aortic smooth muscle cells on the fiber scaffold made from pig gelatin, TG enzyme, and chemical crosslinking agent EDC/NHS via immersion rotation jet spinning technology. It was confirmed that both types of muscle cells adhere to the gelatin fibers which promoted their mature alignment within anisotropic 3D muscle structure [48] (Figure 2b). This provides a new idea for the large-scale production of culture meat.

### 4.3. Extrusion

Extrusion is considered the most widely used technique for converting plant-based materials into fiber products. Texturized vegetable proteins (TVP) are recommended to be utilized in the production of meat analogs through moisture extrusion [68]. TVP mainly produces vegetable protein, and the process involves a series of thermodynamic changes in the material mixture in a reactor, resulting in structures with certain rigidity, hardness, and various complex shapes. After that, the products are obtained for pass processing [69]. In addition to producing a sponge-like structure, their porous pore size helps promote the adequate transport of oxygen, nutrients, and waste [15]. Since the 1960s, research on plant-based proteins had been undertaken to create meat substitutes with texture and taste similar to traditional meat [70]. Nowadays, the demand for TVP in frozen dumplings, ham and sausages, fish balls, and other foods is continually increasing [71], indicating that the world and consumers’ acceptance and satisfaction with TVP products are gradually rising [72].

Proteins, such as soy protein, peanut protein, and gluten, have become crucial components in TVP [73,74]. Among these, the most commonly employed ones are soy flour, soy protein concentrate (SPC), and SPI, which have been discovered to be appropriate materials for promoting cellular adhesion [75]. The highly interconnected sponge-like porous scaffold consisting of soy protein and β-chitosan is used to cultivate mesenchymal stem cells in a 3D environment, indicating excellent biocompatibility and significant potentiality in bio-delivery systems [76]. Another similar study revealed that hybrid cultured meat containing animal cells and plant proteins obtained by covering the coating substrate on porous scaffolds realized myoblast growth and upgraded product texture and sensory characteristics [13] (Figure 3a). While this scaffold technology can support cell adhesion and proliferation, some porous scaffolds may not provide the necessary tissue alignment and sufficient sensory experience, which could pose a drawback for fully cut meat products, and further research is needed.

### 4.4. Directional Freezing

Directional freezing techniques may hold promise to fabricate cultured meat with a specific structure that targets biomaterials with green, fast, and cost-effective features [77]. The technique involves placing colloidal particles or solutions containing biomaterials on one side of a stationary steel plate mold to create a temporary ice template through temperature cooling. The physical cooling causes radial orientation between molds. Ultimately, highly aligned and ordered porous materials are obtained through sublimation using the ice template (template removal) [39,77]. Contrary to porous isotropic scaffolds, directional freezing scaffolds possess both porous and ordered structures that are more conducive to cell alignment, growth, penetration, migration, and oxygen and nutrient exchange [78].

Directional freezing exhibits excellent prospects for material engineering, chemical engineering, and tissue engineering. For example, the ordered porous nanocomposite scaffold, which is constructed by combining chitosan, hydroxyapatite nanoparticles, 3D printing, and directional freezing techniques, displayed exceptional biocompatibility, thereby promoting bone tissue repair and remodeling [78]. In recent years, there have been significant advancements in the food field. Xiang et al. employed a directional freezing technique along with plant protein, wheat gluten, and BSCs to construct a 3D fibronectin scaffold, which was then cultured. The sowing, proliferation, and differentiation of the scaffold were studied to obtain accurate results. The results proved that BSCs successfully migrated, proliferated, and differentiated on the fibronectin scaffold, indicating that the scaffold had the ability to mimic natural muscle fiber tissue and undergo safe meat production [39] (Figure 3b).

### 4.5. Electric Field

In 3D cultures, gel is a common scaffold shape that provides a cell cultivation platform, and it is considered to play a crucial role in cell engineering [79,80]. Electric field technology, with the advantages of easy operation, high efficiency, and low energy consumption, has great potential for application in preparing 3D anisotropic network scaffolds [49,81]. Gels driven by electric fields (E-gel) typically pass a sol–gel transition when exposed to low-voltage direct current (DC) [82]. The anisotropic structural mechanism involves charged molecules transitioning from a disordered state to an ordered state under DC, aligning their dipole moment with an external electric field [83]. Simultaneously, the electrolysis of water causes ions to move towards the opposite electrode, creating and providing a hierarchical structure near the electrode when the local pH is less than the isoelectric point [82]. This technique has been reported more extensively in tissue engineering applications such as bone formation, nerve tissue regeneration, wound healing, and, rarely, in the food industry.

There are some reports about anisotropic scaffolds driven by electric fields. Zeynep analyzed the drug release potential of cultured human keratinocytes in vitro, using a low-voltage electric field to construct silk fibroin gel scaffolds with different proportions of curcumin, and found that the drug release and antibacterial ability can last for 13 days [84]. The combination of SPI and sodium chloride had been studied for fabricating anisotropic hydrogels under electric fields, which exhibited excellent water retention and dissolution resistance, offering a novel approach to developing protein-rich foods for cell and tissue engineering [49] (Figure 4a). In our research, the successful construction of an anisotropic gel scaffold from SPI-bound polysaccharide laid the foundation for the subsequent preparation of cell culture meat (Figure 4b).

### 4.6. Cell Microcarriers

The simplest scaffold technology belongs to cell microcarriers, which are a type of 3D culture scaffold with a diameter of several hundred microns and a large surface/volume ratio. They can wrap functional living cells [51,85], including human mesenchymal stem cells (MSCs) [86], embryonic stem cells (ESCs) [87], and induced pluripotent stem cells (iPSCs) [88], and have been extensively utilized in the vaccine and cell therapy industries to achieve mass cell expansion. Due to their scalability advantage, edible cell microcarriers are being advised for cultured meat, which requires excellent cell affinity and edible biodegradable biomaterials [85]. Specifically, edible materials are made into cell microcarriers, and then can be further embedded and cultured in meat products to improve their texture, taste, and nutritional value [85].

In recent years, reports related to cultured meat are in their infancy. For instance, the objective of this study was to develop edible hydrogel composite MCs using chitosan and collagen for cultivating C2C12 muscle cells, rabbit smooth muscle cells, and sheep fibroblast cells. Results showed that the carrier surface could be fully covered and cells could be attached quickly, leading to rapid proliferation within just a few days [50] (Figure 4c). Additionally, a 3D porous microcarrier prepared for gelatin (PoGelat-MC) was used as a cell expansion scaffold to promote the growth of pig skeletal muscle cells, mouse muscle cells, and mouse adipose cells. After combining with 3D-printed molds and glutamine transaminase, the minced pig muscle tissue was assembled into centimeter-scale meatballs, which established superior mechanical properties and protein content in natural pork meatballs [51]. Norris et al. used oil-in-water emulsion as a template to fabricate edible gelatin microcarriers for culturing C2C12 or BSCs, which can support muscle cell proliferation and differentiation, and the resulting muscle tissue could be cooked into meat patties that maintain their shape and exhibit browning during cooking [52]. These findings suggest that edible microcarriers have immense potential for the growing of future cultured meat products.

### 4.7. Plant Tissue Decellularization

Traditional decellularization technology has initially been employed in the medical industry to address issues like transplantation shortage and in vitro models [89,90]. However, this method involving animal slaughter is a disservice to sustainable development and animal welfare concepts; therefore, a plant tissue decellularization method has been proposed. This approach includes removing cells from plant tissue through a combination of physical (ultrasound, oscillation, and electric shock), chemical (detergents, acids and bases, and organic solvents), and other methods while preserving the tissue’s original structure and system, and then cultivating it in a bioreactor [9,91]. The crucial point that is certain plant tissues with inherent anisotropy, containing celery, spinach, chives, and apples, and their distinctive vascular and porous structures work to overcome the challenge of transporting oxygen and nutrients. Using these materials as cell culture scaffolds can induce cell alignment without other methods [15]. Moreover, vegetable- and fruit-related plants may ignore concerns about edibility and economic value.

The potential for developing cultured meat was tested applying spinach leaves as a decellularized scaffold to foster primary BSCs from three different cows. After 14 days of cultivation and 7 days of differentiation, cells could maintain 99% cell viability and had significantly improved differentiation, respectively. While cell alignment varied to some degree, all exhibited a strong directional arrangement [53] (Figure 5a). Additionally, an example of a 3D scaffold was the combination of decellularized broccoli florets and microcarriers, which increased the nutritional performance of the product and supported the adhesion and viability of BSCs [54]. These studies have identified an economically feasible and easily reproducible pathway that does not contain other animal-derived components, and expands important prospects for future applications.

### 4.8. Cell Sheet Engineering

Three-dimensional printing, electrospinning, extrusion, and directional freezing scaffold technologies all build a three-dimensional biological scaffold to cultivate cells, obtaining a certain tissue structure and ultimately controlling their geometry. Cell sheet engineering (CSE) proposes a novel method of producing tissue-like structures without scaffolding by using cell sheet layering technology. Single- or multilayer sheet-like structures are connected by ECMs secreted by cells themselves. As cell–cell connections and fusion increase in density, a single layer is formed and stacked on the next layer, and assembled into a 3D tissue structure [90,92,93]. In addition, the characteristics of high-density cell bundles can form a dense cellular network, providing new hope for cultivating meat.

Recently, laboratory-scale cultured meat models have been developed. Shahin-Shamsabadi A designed cell sheets for co-culturing mouse and 3T3-L1 (pre-adipocytes) to generate 3D muscle-like structures. The study found that the co-cultured protein and lipid content were greater compared to individually cultured cells, indicating the possibility of creating cultured meat through co-culturing [55]. Hong and his team utilized chitosan–cellulose as raw materials to fabricate a porous nanofilm cell sheet that was used for culturing mouse C2C12 cells. Allophycocyanin was incorporated as a nutritional and transportation platform within the cell sheet, which facilitated cell growth and yielded a muscle-like tissue measuring 2 cm^2^. Cooking the cell sheet through baking or frying produced results similar to Italian sausages [56] (Figure 5b). In summary, based on a simple and effective method, the cell sheet technology does not require additional biomaterials and only cultivates cells on the produced cell sheets, which has the advantages of safety and sustainability. However, scalability, cell arrangement structure, and applicability could pose significant challenges for this technology.

## 5. Potential Benefits and Challenges of the Cultured Meat

Cultivated meat represents an innovative technological revolution that has both potential benefits and uncertainties. Compared to traditional meat, cultured meat has the potential to satisfy the constant demand from consumers for meat, and promote sustainable development by gradually reducing the land and water resources occupied by traditional meat farming [12]. The greenhouse gas emissions and quantity of grain feed required have decreased slightly, making cultivated meat an environmentally friendly alternative [3,12]. Moreover, this technology can lower the risk of foodborne illnesses and biological risks, resulting in healthier products [94]. Additionally, more sustainable raw materials, such as proteins or polysaccharides from plant or microbial sources, can be used for the production of cultured meat [95].

However, challenges exist in the preparation of cultured meat at this stage. The primary challenge is the dependence on expensive scaffold equipment, raw materials (stem cell sources and scaffold biomaterials), and serum medium [15,94]. Upgrading these factors is necessary. Additionally, there is still a significant difference in texture, flavor, taste, and nutritional value when compared to farmed meat [36]. Furthermore, long-term edible safety and regulatory mechanisms for large-scale demand require attention [95,96]. We believe that these challenges posed by scaffold technology and serum-free culture media can foster industrial development of cultured meat through continuous reform and innovation. In conclusion, we provide a table (Table 3) comparing the advantages and disadvantages of cultured meat and farmed meat.

## 6. Conclusions and Prospects

Cultured meat is a promising solution for preparing safer and healthier food products that facilitate sustainable development in the future. The introduction of 3D scaffold technologies has proven highly advantageous in constructing products with superior exterior structure and functional applicability. Nowadays, 3D printing and electrospinning have already developed in tissue engineering for wound healing and cartilage regeneration. To establish 3D scaffolds for cultured meat, there are several important points to consider. Viable biomaterials and cells, namely, muscle and fat cells, must be explored in combination with scaffold technologies to ensure biocompatibility and support cell functionality. Furthermore, safety and biodegradability should be taken into account. To guarantee scale production and consumer acceptance, compliance with green and sustainable objectives is also essential.

The optimization of scaffolding processes, cost reduction, and the construction of comprehensive and diverse cultured meats are necessary, including considerations such as scaffold strength, co-culture of multiple cells, product texture, flavor, and nutritional value. It also involves cell selection and medium formulation such as partial replacement or serum-free culture. Ultimately, research must continue towards green, simple, and sustainable production methods to move commercialization. As the technology expands and matures, media coverage and businesses should implement more education and popularization to rapidly integrate into the market. Relevant regulatory and legislative departments should strengthen measures to assure the safety of cultured meat. Furthermore, it is crucial to enhance public awareness of environmental preservation and sustainability by means of media and other popular scientific promotional channels. We believe that cultured meat provides a new option for sustainable and animal welfare benefits. In brief, the technique of cultured meat aims to alleviate the strain caused by traditional meat production. Nevertheless, it cannot completely replace animal husbandry. Moving forward, the symbiosis of cultured and farmed meat could bring advantageous outcomes for consumers and our planet.

## Figures and Tables

**Figure 1 foods-12-02614-f001:**
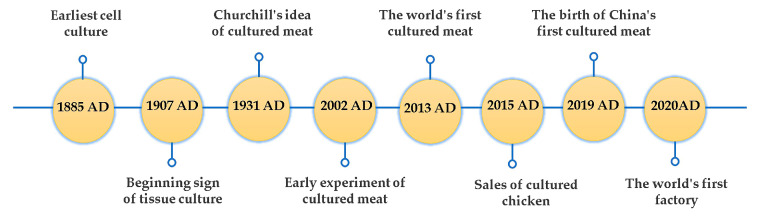
Timeline of the history of cultured meat.

**Figure 2 foods-12-02614-f002:**
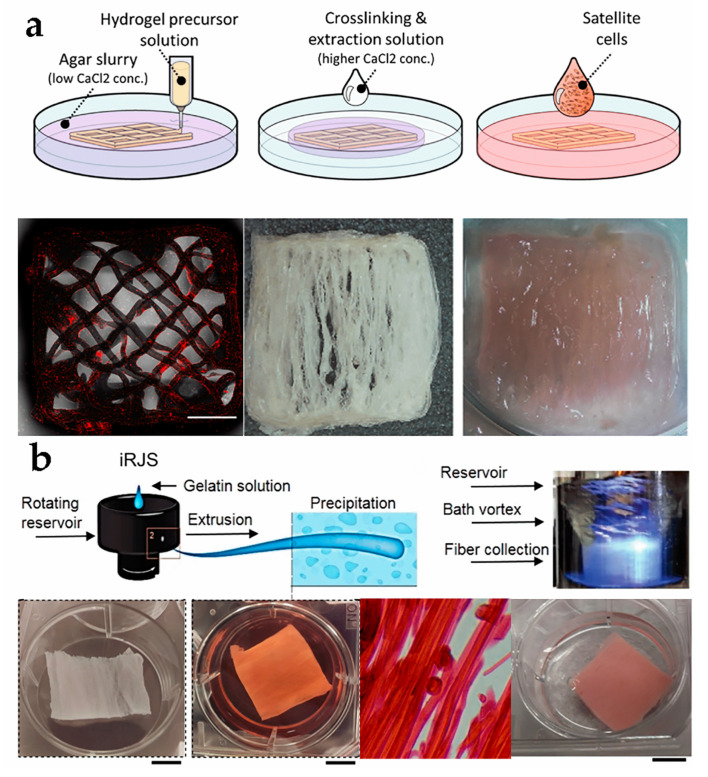
(**a**) Process diagram of using 3D printing to create scaffold and cultivate cells into tissue [46] (reprinted from lanovici et al. (2022) with permission from Elsevier Ltd.). (**b**) Schematic diagram of electrospinning and tissue culture [48] (adapted from MacQueen et al. (2019) with permission from Springer Nature).

**Figure 3 foods-12-02614-f003:**
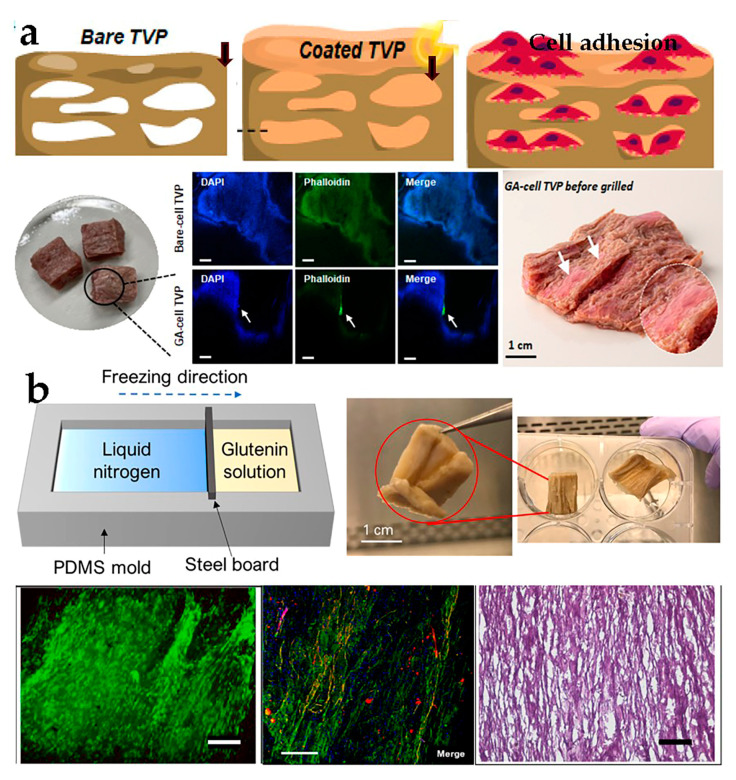
(**a**) Construction of cultivated meat with combined TVP coating substrate [13] (adapted from Lee et al. (2022) with copyright from American Chemical Society). (**b**) Directional freezing form process and cell proliferation and differentiation [39] (adapted from Xiang et al. (2022) with permission from Elsevier Ltd.).

**Figure 4 foods-12-02614-f004:**
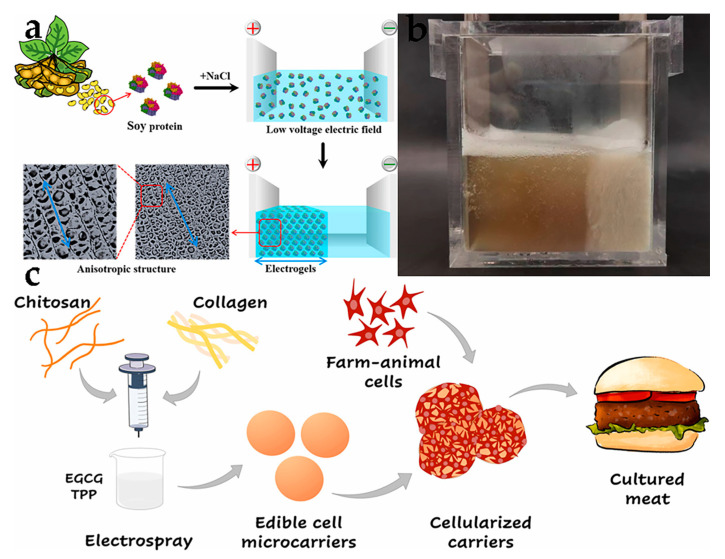
(**a**,**b**) Anisotropic scaffolds driven by electric field and plant protein [49] (reprinted from Cao et al. (2023) from Elsevier Ltd.). (**c**) The vector diagrams of cultured meat obtained through microcarriers [50] (reprinted from Zernov et al. (2022) with permission from Elsevier Ltd.).

**Figure 5 foods-12-02614-f005:**
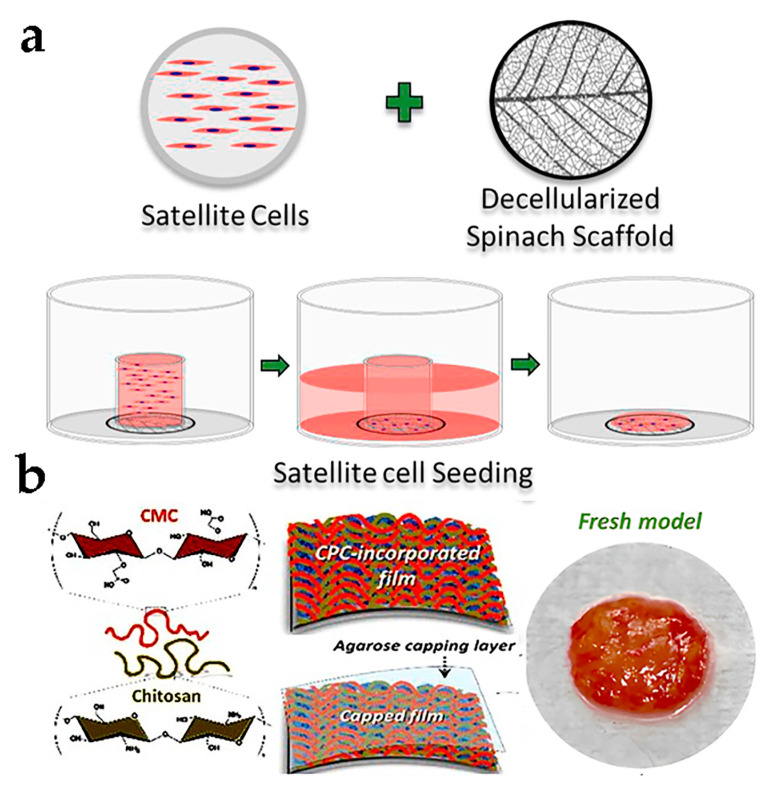
(**a**) Formation progress of BSC attachment on decellularized scaffold [53] (reprinted from Jones et al. (2021) with permission from Elsevier Ltd.). (**b**) Constitution of model meat cell sheets including material preparation, serum substitution, and product display [56] (reprinted from Park et al. (2021) with copyright from American Chemical Society).

**Table 1 foods-12-02614-t001:** Advantages and disadvantages of various scaffold biomaterials.

Types of Scaffold Biomaterials	Advantages	Disadvantages
Animal-derived biomaterials	Excellent biocompatibility Promoting cell growth Nutrition rich	Relatively high production costs Environmental burden Unfriendly animal welfare
Plant-derived biomaterials	Healthy Sustainable Favorable nutritional value Natural 3D structure Relatively low cost Safety	Inferior scalability Poor tissue applicability
Synthetic polymer biomaterials	Multiple structures Excellent mechanical features	Potential toxicity Low nutritional value Slow degradation rate

**Table 2 foods-12-02614-t002:** Summary of the biomaterials and cell types involved in the cultured meat using various scaffold technologies.

Scaffold Technologies	Biomaterials	Cultured Cells	Innovative Points in Cultivating Meat	References
3D printing	Plant protein, edible polysaccharides (seaweed salts, gelatin)	Bovine satellite cells, adipose-derived stem cells	Printing out ideal thickness and striped cultured meat products	[46,47]
Electrospinning	Gelatin, TG enzyme	Rabbit skeletal muscle cells, bovine aortic smooth muscle cells	Cultivated meat with certain fiber filaments and anisotropic structure	[48]
Extrusion	Soy protein	C2C12 muscle cells	Porous structure and texture similar to traditional meat	[13]
Directional freezing	Wheat gluten	BSCs	Anisotropic structure and mimic natural muscle fiber tissue	[39]
Electric field	Soy protein, polysaccharide	No cells involved	Anisotropic structure similar to traditional meat	[49]
Cell microcarriers	Chitosan, collagen, gelatin	C2C12, BSCs, rabbit smooth muscle cells, pig skeletal muscle cells	Similar sensory characteristics and nutritional value to traditional meat	[50,51,52]
Plant tissue decellularization	Natural plant tissue (spinach, broccoli floret)	BSCs	Natural plant vessels provide directional arrangement of cultured meat	[53,54]
Cell sheet engineering	Chitosan, cellulose	C2C12, 3T3-L1	Cultivated meat with a multi-layer thickness and rich nutrition	[55,56]

**Table 3 foods-12-02614-t003:** Advantages and disadvantages of cultured meat versus livestock meat.

Types of Meat	Advantages	Disadvantages
Cultured meat	Environmental resource efficiency Health and sustainability Non-animal protein utilization	Expensive production process Unsatisfied food sensory Lack of policies and regulations Challenge of large-scale production
Livestock meat	High nutritional value Better energy and mood Easy consumer acceptance	Resource unsustainability Health risks Potentially ethical issues

## Data Availability

Not applicable.

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
