# Peer review of "An Overview of Recent Progress in Engineering Three-Dimensional Scaffolds for Cultured Meat Production"

_foods, 2023, doi:10.3390/foods12132614_

Round 1
Reviewer 1 Report
The manuscript foods-2465674 entitled " An Overview of Recent Progress in Engineering Three-Dimensional Scaffolds for Cultured Meat Production”
Overall, the paper is well written and very well explained.
I have few observations that need to be addressed to improve the paper’s quality.
There is too much data in the manuscript. I recommend to construct 1 or 2 tables providing all the information for easy understanding of the readers.
I am suggesting a recent relevant review for reading.
“Prospects of artificial meat: Opportunities and challenges around consumer acceptance”.
Please discuss the consumer acceptance possibilities in the introduction section.
Please discuss the importance of this technique to fulfil the increased demands of the growing population of the world. I recommend to introduce this technique to support the livestock industry not to replace the livestock sector. Please delete the sentence of animal cruelty might be the reason to develop cultured meat.
Also, discuss the advantages and disadvantages of the cultured meat over livestock meat products (real) as a separate section
Author Response
Dear Reviewer,
Thanks for the reviewer’s comments. We have revised our manuscript point by point according to your suggestion. Please see the attachment.

Reviewer 2 Report
Line 12: please check the sentence “the products have been prepared lack sufficient….”
Line 18: please, check the sentence “for the biomanufacturer of cultured.”
Line 27: after “….United Nations” write “(FAO)”
Lines 32-33: “Billions of animals……..intense influence”. This sentence does not seem to be related to the previously expressed ideas. Please, rephrase the sentence.
Line 194: “Derived from the various…..”; you can replace “derived” with another word
Line 213: “Ianovici et al”
Line 251: The Figure 3b: part a) is fully copied from the figure reported by Li et al., 2021.Part b) is fully copied from the figure reported by MacQueen et al.(2019).
The figure 4a is fully copied from the figure reported by Las Heras et al. (2020).
The figure 4b is fully copied from the figure reported by Lee et al. (2022)
The figure 5b is fully copied from the figure reported by Xiang et al. (2022)
The figure 5a is fully copied from the figure reported by Jiang et al. (2022)
Is the figure 6a fully copied from the figure reported by KarahaliloÄŸlu (2018)?
The Figure 7b is fully copied from the figure reported by Liu et al. (2022)
The Figure 7c is fully copied from the figure reported by Norris et al. (2022)
The Figure 8a is fully copied from the figure reported by Jones et al.(2021)
The Figure 8b is fully copied from the figure reported by Thyden et al. (2022)
Is the figure 9a fully copied from the figure reported by Shahin-Shamsabadi and Selvaganapathy (2022)?
The Figure 9b is fully copied from the figure reported by Park et al. (2021)- alongside the ABSTRACT
In my view, that is not ethically correct. The authors should eliminate both figures.
The review should, therefore, be improved by further scientific information.
Reference 43: (CAC), C.A.C. General Standard for Food Additives.
Please, report as Codex Alimentarius Commission (CAC). General Standard for Food Additives. CODEX STAN 192-2019 ---The last revision dates back to 2021
References: please, follow the “Instructions for Authors”
English need minor revision
Author Response

(The authors gave the same response as above.)
